# Deep Learning Face Representation by Joint Identification-Verification

**Yi Sun**[1]     **Yuheng Chen**[2]     **Xiaogang Wang**[3,4]     **Xiaoou Tang**[1,4]
[1]Department of Information Engineering, The Chinese University of Hong Kong
[2]SenseTime Group
[3]Department of Electronic Engineering, The Chinese University of Hong Kong
[4]Shenzhen Institutes of Advanced Technology, Chinese Academy of Sciences
sy011@ie.cuhk.edu.hk   chyh1990@gmail.com
xgwang@ee.cuhk.edu.hk   xtang@ie.cuhk.edu.hk

## Abstract

The key challenge of face recognition is to develop effective feature representations for reducing intra-personal variations while enlarging inter-personal differences. In this paper, we show that it can be well solved with deep learning and using both face identification and verification signals as supervision. The Deep IDentification-verification features (DeepID2) are learned with carefully designed deep convolutional networks. The face identification task increases the inter-personal variations by drawing DeepID2 features extracted from different identities apart, while the face verification task reduces the intra-personal variations by pulling DeepID2 features extracted from the same identity together, both of which are essential to face recognition. The learned DeepID2 features can be well generalized to new identities unseen in the training data. On the challenging LFW dataset [11], **99.15**% face verification accuracy is achieved. Compared with the best previous deep learning result [20] on LFW, the error rate has been significantly reduced by **67**%.

## 1   Introduction

Faces of the same identity could look much different when presented in different poses, illuminations, expressions, ages, and occlusions. Such variations within the same identity could overwhelm the variations due to identity differences and make face recognition challenging, especially in unconstrained conditions. Therefore, reducing the intra-personal variations while enlarging the inter-personal differences is a central topic in face recognition. It can be traced back to early subspace face recognition methods such as LDA [1], Bayesian face [16], and unified subspace [22, 23]. For example, LDA approximates inter- and intra-personal face variations by using two scatter matrices and finds the projection directions to maximize the ratio between them. More recent studies have also targeted the same goal, either explicitly or implicitly. For example, metric learning [6, 9, 14] maps faces to some feature representation such that faces of the same identity are close to each other while those of different identities stay apart. However, these models are much limited by their linear nature or shallow structures, while inter- and intra-personal variations are complex, highly nonlinear, and observed in high-dimensional image space.

In this work, we show that deep learning provides much more powerful tools to handle the two types of variations. Thanks to its deep architecture and large learning capacity, effective features for face recognition can be learned through hierarchical nonlinear mappings. We argue that it is essential to learn such features by using two supervisory signals simultaneously, i.e. the face identification and verification signals, and the learned features are referred to as Deep IDentification-verification features (DeepID2). Identification is to classify an input image into a large number of identity

classes, while verification is to classify a pair of images as belonging to the same identity or not (i.e. binary classification). In the training stage, given an input face image with the identification signal, its DeepID2 features are extracted in the top hidden layer of the learned hierarchical nonlinear feature representation, and then mapped to one of a large number of identities through another function $g$(DeepID2). In the testing stage, the learned DeepID2 features can be generalized to other tasks (such as face verification) and new identities unseen in the training data. The identification supervisory signal tends to pull apart the DeepID2 features of different identities since they have to be classified into different classes. Therefore, the learned features would have rich identity-related or inter-personal variations. However, the identification signal has a relatively weak constraint on DeepID2 features extracted from the same identity, since dissimilar DeepID2 features could be mapped to the same identity through function $g(\cdot)$. This leads to problems when DeepID2 features are generalized to new tasks and new identities in test where $g$ is not applicable anymore. We solve this by using an additional face verification signal, which requires that every two DeepID2 feature vectors extracted from the same identity are close to each other while those extracted from different identities are kept away. The strong per-element constraint on DeepID2 features can effectively reduce the intra-personal variations. On the other hand, using the verification signal alone (i.e. only distinguishing a pair of DeepID2 feature vectors at a time) is not as effective in extracting identity-related features as using the identification signal (i.e. distinguishing thousands of identities at a time). Therefore, the two supervisory signals emphasize different aspects in feature learning and should be employed together.

To characterize faces from different aspects, complementary DeepID2 features are extracted from various face regions and resolutions, and are concatenated to form the final feature representation after PCA dimension reduction. Since the learned DeepID2 features are diverse among different identities while consistent within the same identity, it makes the following face recognition easier. Using the learned feature representation and a recently proposed face verification model [3], we achieved the highest **99.15%** face verification accuracy on the challenging and extensively studied LFW dataset [11]. This is the first time that a machine provided with only the face region achieves an accuracy on par with the $99.20\%$ accuracy of human to whom the entire LFW face image including the face region and large background area are presented to verify.

In recent years, a great deal of efforts have been made for face recognition with deep learning [5, 10, 18, 26, 8, 21, 20, 27]. Among the deep learning works, [5, 18, 8] learned features or deep metrics with the verification signal, while DeepFace [21] and our previous work DeepID [20] learned features with the identification signal and achieved accuracies around $97.45\%$ on LFW. Our approach significantly improves the state-of-the-art. The idea of jointly solving the classification and verification tasks was applied to general object recognition [15], with the focus on improving classification accuracy on fixed object classes instead of hidden feature representations. Our work targets on learning features which can be well generalized to new classes (identities) and the verification task.

## 2 Identification-verification guided deep feature learning

We learn features with variations of deep convolutional neural networks (deep ConvNets) [12]. The convolution and pooling operations in deep ConvNets are specially designed to extract visual features hierarchically, from local low-level features to global high-level ones. Our deep ConvNets take similar structures as in [20]. It contains four convolutional layers, with local weight sharing [10] in the third and fourth convolutional layers. The ConvNet extracts a 160-dimensional DeepID2 feature vector at its last layer (DeepID2 layer) of the feature extraction cascade. The DeepID2 layer to be learned are fully-connected to both the third and fourth convolutional layers. We use rectified linear units (ReLU) [17] for neurons in the convolutional layers and the DeepID2 layer. An illustration of the ConvNet structure used to extract DeepID2 features is shown in Fig. 1 given an RGB input of size $55 \times 47$. When the size of the input region changes, the map sizes in the following layers will change accordingly. The DeepID2 feature extraction process is denoted as $f = \mathrm{Conv}(x, \theta_c)$, where $\mathrm{Conv}(\cdot)$ is the feature extraction function defined by the ConvNet, $x$ is the input face patch, $f$ is the extracted DeepID2 feature vector, and $\theta_c$ denotes ConvNet parameters to be learned.

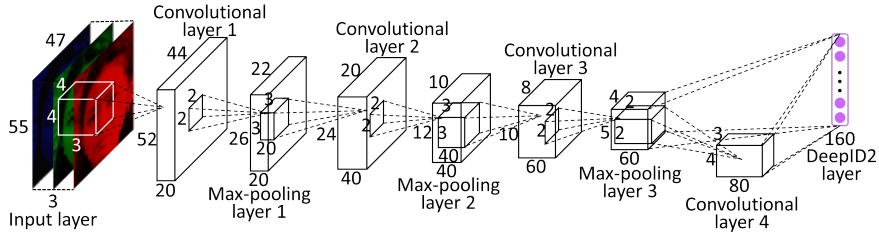

Figure 1: The ConvNet structure for DeepID2 feature extraction.

DeepID2 features are learned with two supervisory signals. The first is face identification signal, which classifies each face image into one of $n$ (e.g., $n = 8192$) different identities. Identification is achieved by following the DeepID2 layer with an $n$-way softmax layer, which outputs a probability distribution over the $n$ classes. The network is trained to minimize the cross-entropy loss, which we call the identification loss. It is denoted as

$$\text{Ident}(f, t, \theta_{id}) = -\sum_{i=1}^{n} p_i \log \hat{p}_i = -\log \hat{p}_t \,, \tag{1}$$

where $f$ is the DeepID2 feature vector, $t$ is the target class, and $\theta_{id}$ denotes the softmax layer parameters. $p_i$ is the target probability distribution, where $p_i = 0$ for all $i$ except $p_t = 1$ for the target class $t$. $\hat{p}_i$ is the predicted probability distribution. To correctly classify all the classes simultaneously, the DeepID2 layer must form discriminative identity-related features (i.e. features with large inter-personal variations). The second is face verification signal, which encourages DeepID2 features extracted from faces of the same identity to be similar. The verification signal directly regularize DeepID2 features and can effectively reduce the intra-personal variations. Commonly used constraints include the L1/L2 norm and cosine similarity. We adopt the following loss function based on the L2 norm, which was originally proposed by Hadsell et al.[7] for dimensionality reduction,

$$\text{Verif}(f_i, f_j, y_{ij}, \theta_{ve}) = \begin{cases} \frac{1}{2} \left\| f_i - f_j \right\|_2^2 & \text{if } y_{ij} = 1 \\ \frac{1}{2} \max \left( 0, m - \left\| f_i - f_j \right\|_2 \right)^2 & \text{if } y_{ij} = -1 \end{cases} \,, \tag{2}$$

where $f_i$ and $f_j$ are DeepID2 feature vectors extracted from the two face images in comparison. $y_{ij} = 1$ means that $f_i$ and $f_j$ are from the same identity. In this case, it minimizes the L2 distance between the two DeepID2 feature vectors. $y_{ij} = -1$ means different identities, and Eq. (2) requires the distance larger than a margin $m$. $\theta_{ve} = \{m\}$ is the parameter to be learned in the verification loss function. Loss functions based on the L1 norm could have similar formulations [15]. The cosine similarity was used in [17] as

$$\text{Verif}(f_i, f_j, y_{ij}, \theta_{ve}) = \frac{1}{2} \left( y_{ij} - \sigma(wd + b) \right)^2 \,, \tag{3}$$

where $d = \frac{f_i \cdot f_j}{\|f_i\|_2 \|f_j\|_2}$ is the cosine similarity between DeepID2 feature vectors, $\theta_{ve} = \{w, b\}$ are learnable scaling and shifting parameters, $\sigma$ is the sigmoid function, and $y_{ij}$ is the binary target of whether the two compared face images belong to the same identity. All the three loss functions are evaluated and compared in our experiments.

Our goal is to learn the parameters $\theta_c$ in the feature extraction function $\text{Conv}(\cdot)$, while $\theta_{id}$ and $\theta_{ve}$ are only parameters introduced to propagate the identification and verification signals during training. In the testing stage, only $\theta_c$ is used for feature extraction. The parameters are updated by stochastic gradient descent. The identification and verification gradients are weighted by a hyperparameter $\lambda$. Our learning algorithm is summarized in Tab. 1. The margin $m$ in Eq. (2) is a special case, which cannot be updated by gradient descent since this will collapse it to zero. Instead, $m$ is fixed and updated every $N$ training pairs ($N \approx 200,000$ in our experiments) such that it is the threshold of

Table 1: The DeepID2 feature learning algorithm.

---

**input**: training set $\chi = \{(x_i, l_i)\}$, initialized parameters $\theta_c$, $\theta_{id}$, and $\theta_{ve}$, hyperparameter $\lambda$, learning rate $\eta(t)$, $t \leftarrow 0$

---

**while** not converge **do**
$\quad t \leftarrow t + 1 \quad$ sample two training samples $(x_i, l_i)$ and $(x_j, l_j)$ from $\chi$
$\quad f_i = \mathrm{Conv}(x_i, \theta_c)$ and $f_j = \mathrm{Conv}(x_j, \theta_c)$
$\quad \nabla\theta_{id} = \frac{\partial \mathrm{Ident}(f_i, l_i, \theta_{id})}{\partial \theta_{id}} + \frac{\partial \mathrm{Ident}(f_j, l_j, \theta_{id})}{\partial \theta_{id}}$
$\quad \nabla\theta_{ve} = \lambda \cdot \frac{\partial \mathrm{Verif}(f_i, f_j, y_{ij}, \theta_{ve})}{\partial \theta_{ve}}$, where $y_{ij} = 1$ if $l_i = l_j$, and $y_{ij} = -1$ otherwise.
$\quad \nabla f_i = \frac{\partial \mathrm{Ident}(f_i, l_i, \theta_{id})}{\partial f_i} + \lambda \cdot \frac{\partial \mathrm{Verif}(f_i, f_j, y_{ij}, \theta_{ve})}{\partial f_i}$
$\quad \nabla f_j = \frac{\partial \mathrm{Ident}(f_j, l_j, \theta_{id})}{\partial f_j} + \lambda \cdot \frac{\partial \mathrm{Verif}(f_i, f_j, y_{ij}, \theta_{ve})}{\partial f_j}$
$\quad \nabla\theta_c = \nabla f_i \cdot \frac{\partial \mathrm{Conv}(x_i, \theta_c)}{\partial \theta_c} + \nabla f_j \cdot \frac{\partial \mathrm{Conv}(x_j, \theta_c)}{\partial \theta_c}$
$\quad$ update $\theta_{id} = \theta_{id} - \eta(t) \cdot \nabla\theta_{id}$, $\theta_{ve} = \theta_{ve} - \eta(t) \cdot \nabla\theta_{ve}$, and $\theta_c = \theta_c - \eta(t) \cdot \nabla\theta_c$.
**end while**
**output** $\theta_c$

---

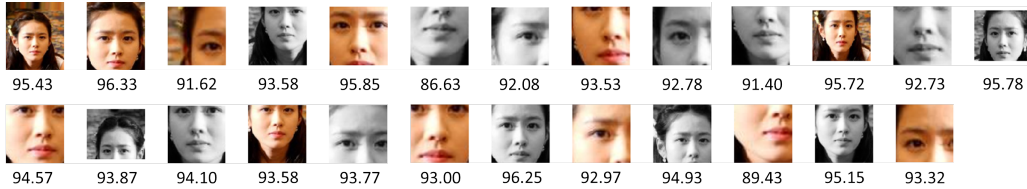

| 95.43 | 96.33 | 91.62 | 93.58 | 95.85 | 86.63 | 92.08 | 93.53 | 92.78 | 91.40 | 95.72 | 92.73 | 95.78 |

| 94.57 | 93.87 | 94.10 | 93.58 | 93.77 | 93.00 | 96.25 | 92.97 | 94.93 | 89.43 | 95.15 | 93.32 |

Figure 2: Patches selected for feature extraction. The Joint Bayesian [3] face verification accuracy (%) using features extracted from each individual patch is shown below.

the feature distances $\|f_i - f_j\|$ to minimize the verification error of the previous N training pairs. Updating $m$ is not included in Tab. 1 for simplicity.

## 3 Face Verification

To evaluate the feature learning algorithm described in Sec. 2, DeepID2 features are embedded into the conventional face verification pipeline of face alignment, feature extraction, and face verification. We first use the recently proposed SDM algorithm [24] to detect 21 facial landmarks. Then the face images are globally aligned by similarity transformation according to the detected landmarks. We cropped 400 face patches, which vary in positions, scales, color channels, and horizontal flipping, according to the globally aligned faces and the position of the facial landmarks. Accordingly, 400 DeepID2 feature vectors are extracted by a total of 200 deep ConvNets, each of which is trained to extract two 160-dimensional DeepID2 feature vectors on one particular face patch and its horizontally flipped counterpart, respectively, of each face.

To reduce the redundancy among the large number of DeepID2 features and make our system practical, we use the forward-backward greedy algorithm [25] to select a small number of effective and complementary DeepID2 feature vectors (25 in our experiment), which saves most of the feature extraction time during test. Fig. 2 shows all the selected 25 patches, from which 25 160-dimensional DeepID2 feature vectors are extracted and are concatenated to a 4000-dimensional DeepID2 feature vector. The 4000-dimensional vector is further compressed to 180 dimensions by PCA for face verification. We learned the Joint Bayesian model [3] for face verification based on the extracted DeepID2 features. Joint Bayesian has been successfully used to model the joint probability of two faces being the same or different persons [3, 4].

# 4 Experiments

We report face verification results on the LFW dataset [11], which is the de facto standard test set for face verification in unconstrained conditions. It contains $13,233$ face images of $5749$ identities collected from the Internet. For comparison purposes, algorithms typically report the mean face verification accuracy and the ROC curve on $6000$ given face pairs in LFW. Though being sound as a test set, it is inadequate for training, since the majority of identities in LFW have only one face image. Therefore, we rely on a larger outside dataset for training, as did by all recent high-performance face verification algorithms [4, 2, 21, 20, 13]. In particular, we use the CelebFaces+ dataset [20] for training, which contains $202,599$ face images of $10,177$ identities (celebrities) collected from the Internet. People in CelebFaces+ and LFW are mutually exclusive. DeepID2 features are learned from the face images of $8192$ identities randomly sampled from CelebFaces+ (referred to as CelebFaces+A), while the remaining face images of $1985$ identities (referred to as CelebFaces+B) are used for the following feature selection and learning the face verification models (Joint Bayesian). When learning DeepID2 features on CelebFaces+A, CelebFaces+B is used as a validation set to decide the learning rate, training epochs, and hyperparameter $\lambda$. After that, CelebFaces+B is separated into a training set of $1485$ identities and a validation set of $500$ identities for feature selection. Finally, we train the Joint Bayesian model on the entire CelebFaces+B data and test on LFW using the selected DeepID2 features. We first evaluate various aspect of feature learning from Sec. 4.1 to Sec. 4.3 by using a single deep ConvNet to extract DeepID2 features from the entire face region. Then the final system is constructed and compared with existing best performing methods in Sec. 4.4.

## 4.1 Balancing the identification and verification signals

We investigates the interactions of identification and verification signals on feature learning, by varying $\lambda$ from $0$ to $+\infty$. At $\lambda = 0$, the verification signal vanishes and only the identification signal takes effect. When $\lambda$ increases, the verification signal gradually dominates the training process. At the other extreme of $\lambda \rightarrow +\infty$, only the verification signal remains. The L2 norm verification loss in Eq. (2) is used for training. Figure 3 shows the face verification accuracy on the test set by comparing the learned DeepID2 features with L2 norm and the Joint Bayesian model, respectively. It clearly shows that neither the identification nor the verification signal is the optimal one to learn features. Instead, effective features come from the appropriate combination of the two.

This phenomenon can be explained from the view of inter- and intra-personal variations, which could be approximated by LDA. According to LDA, the inter-personal scatter matrix is $S_{inter} = \sum_{i=1}^{c} n_i \cdot (\bar{x}_i - \bar{x})(\bar{x}_i - \bar{x})^\top$, where $\bar{x}_i$ is the mean feature of the $i$-th identity, $\bar{x}$ is the mean of the entire dataset, and $n_i$ is the number of face images of the $i$-th identity. The intra-personal scatter matrix is $S_{intra} = \sum_{i=1}^{c} \sum_{x \in D_i} (x - \bar{x}_i)(x - \bar{x}_i)^\top$, where $D_i$ is the set of features of the $i$-th identity, $\bar{x}_i$ is the corresponding mean, and $c$ is the number of different identities. The inter- and intra-personal variances are the eigenvalues of the corresponding scatter matrices, and are shown in Fig. 5. The corresponding eigenvectors represent different variation patterns. Both the magnitude and diversity of feature variances matter in recognition. If all the feature variances concentrate on a small number of eigenvectors, it indicates the diversity of intra- or inter-personal variations is low. The features are learned with $\lambda = 0$, $0.05$, and $+\infty$, respectively. The feature variances of each given $\lambda$ are normalized by the corresponding mean feature variance.

When only the identification signal is used ($\lambda = 0$), the learned features contain both diverse inter- and intra-personal variations, as shown by the long tails of the red curves in both figures. While diverse inter-personal variations help to distinguish different identities, large and diverse intra-personal variations are disturbing factors and make face verification difficult. When both the identification and verification signals are used with appropriate weighting ($\lambda = 0.05$), the diversity of the inter-personal variations keeps unchanged while the variations in a few main directions become even larger, as shown by the green curve in the left compared to the red one. At the same time, the intra-personal variations decrease in both the diversity and magnitude, as shown by the green curve in the right. Therefore, both the inter- and intra-personal variations changes in a direction that makes face verification easier. When $\lambda$ further increases towards infinity, both the inter- and intra-personal variations collapse to the variations in only a few main directions, since without the identification signal, diverse features cannot be formed. With low diversity on inter-

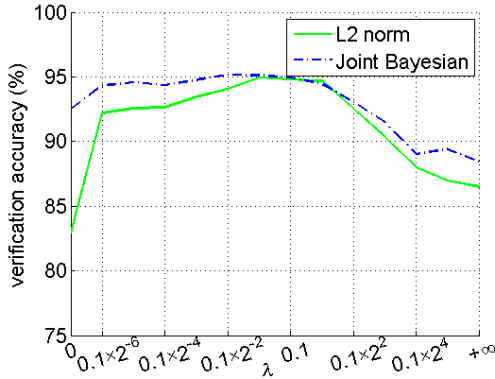

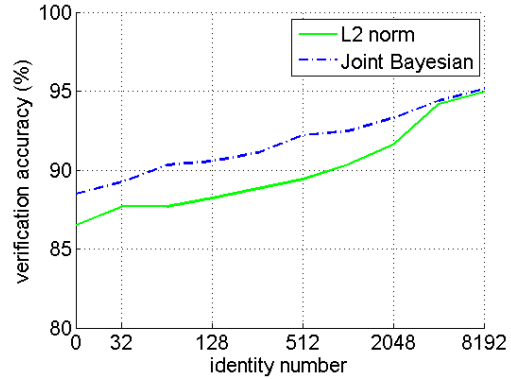

Figure 3: Face verification accuracy by varying the weighting parameter $\lambda$. $\lambda$ is plotted in log scale.

Figure 4: Face verification accuracy of DeepID2 features learned by both the the face identification and verification signals, where the number of training identities (shown in log scale) used for face identification varies. The result may be further improved with more than $8192$ identities.

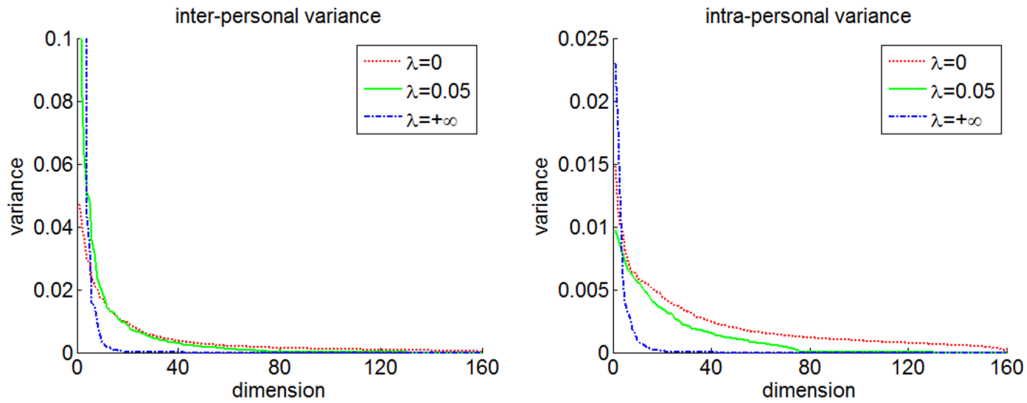

Figure 5: Spectrum of eigenvalues of the inter- and intra-personal scatter matrices. Best viewed in color.

personal variations, distinguishing different identities becomes difficult. Therefore the performance degrades significantly.

Figure 6 shows the first two PCA dimensions of features learned with $\lambda = 0$, $0.05$, and $+\infty$, respectively. These features come from the six identities with the largest numbers of face images in LFW, and are marked by different colors. The figure further verifies our observations. When $\lambda = 0$ (left), different clusters are mixed together due to the large intra-personal variations, although the cluster centers are actually different. When $\lambda$ increases to $0.05$ (middle), intra-personal variations are significantly reduced and the clusters become distinguishable. When $\lambda$ further increases towards infinity (right), although the intra-personal variations further decrease, the cluster centers also begin to collapse and some clusters become significantly overlapped (as the red, blue, and cyan clusters in Fig. 6 right), making it hard to distinguish again.

## 4.2 Rich identity information improves feature learning

We investigate how would the identity information contained in the identification supervisory signal influence the learned features. In particular, we experiment with an exponentially increasing number of identities used for identification during training from $32$ to $8192$, while the verification signal is generated from all the $8192$ training identities all the time. Fig. 4 shows how the verification accuracies of the learned DeepID2 features (derived from the L2 norm and Joint Bayesian) vary on the test set with the number of identities used in the identification signal. It shows that

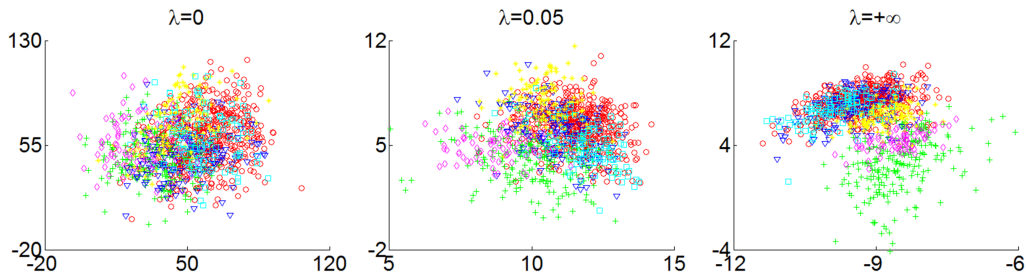

Figure 6: The first two PCA dimensions of DeepID2 features extracted from six identities in LFW.

Table 2: Comparison of different verification signals.

| verification signal | L2 | L2+ | L2- | L1 | cosine | none |
|---|---|---|---|---|---|---|
| L2 norm (%) | 94.95 | 94.43 | 86.23 | 92.92 | 87.07 | 86.43 |
| Joint Bayesian (%) | 95.12 | 94.87 | 92.98 | 94.13 | 93.38 | 92.73 |

identifying a large number (e.g., 8192) of identities is key to learning effective DeepID2 feature representation. This observation is consistent with those in Sec. 4.1. The increasing number of identities provides richer identity information and helps to form DeepID2 features with diverse inter-personal variations, making the class centers of different identities more distinguishable.

### 4.3   Investigating the verification signals

As shown in Sec. 4.1, the verification signal with moderate intensity mainly takes the effect of reducing the intra-personal variations. To further verify this, we compare our L2 norm verification signal on all the sample pairs with those only constrain either the positive or negative sample pairs, denoted as L2+ and L2-, respectively. That is, the L2+ only decreases the distances between DeepID2 features of the same identity, while L2- only increases the distances between DeepID2 features of different identities if they are smaller than the margin. The face verification accuracies of the learned DeepID2 features on the test set, measured by the L2 norm and Joint Bayesian respectively, are shown in Table 2. It also compares with the L1 norm and cosine verification signals, as well as no verification signal (none). The identification signal is the same (classifying the 8192 identities) for all the comparisons.

DeepID2 features learned with the L2+ verification signal are only slightly worse than those learned with L2. In contrast, the L2- verification signal helps little in feature learning and gives almost the same result as no verification signal is used. This is a strong evidence that the effect of the verification signal is mainly reducing the intra-personal variations. Another observation is that the face verification accuracy improves in general whenever the verification signal is added in addition to the identification signal. However, the L2 norm is better than the other compared verification metrics. This may be due to that all the other constraints are weaker than L2 and less effective in reducing the intra-personal variations. For example, the cosine similarity only constrains the angle, but not the magnitude.

### 4.4   Final system and comparison with other methods

Before learning Joint Bayesian, DeepID2 features are first projected to 180 dimensions by PCA. After PCA, the Joint Bayesian model is trained on the entire CelebFaces+B data and tested on the 6000 given face pairs in LFW, where the log-likelihood ratio given by Joint Bayesian is compared to a threshold optimized on the training data for face verification. Tab. 3 shows the face verification accuracy with an increasing number of face patches to extract DeepID2 features, as well as the time used to extract those DeepID2 features from each face with a single Titan GPU. We achieve **98.97**% accuracy with all the 25 selected face patches. The feature extraction process is also efficient and takes only 35 ms for each face image. The face verification accuracy of each individual face patch is provided in Fig. 2. The short DeepID2 signature is extremely efficient for face identification and face image search when matching a query image with a large number of candidates.

Table 3: Face verification accuracy with DeepID2 features extracted from an increasing number of face patches.

| # patches | 1 | 2 | 4 | 8 | 16 | 25 |
|---|---|---|---|---|---|---|
| accuracy (%) | 95.43 | 97.28 | 97.75 | 98.55 | 98.93 | 98.97 |
| time (ms) | 1.7 | 3.4 | 6.1 | 11 | 23 | 35 |

Table 4: Accuracy comparison with the previous best results on LFW.

| method | accuracy (%) |
|---|---|
| High-dim LBP [4] | $95.17 \pm 1.13$ |
| TL Joint Bayesian [2] | $96.33 \pm 1.08$ |
| DeepFace [21] | $97.35 \pm 0.25$ |
| DeepID [20] | $97.45 \pm 0.26$ |
| GaussianFace [13] | $98.52 \pm 0.66$ |
| DeepID2 | $99.15 \pm 0.13$ |

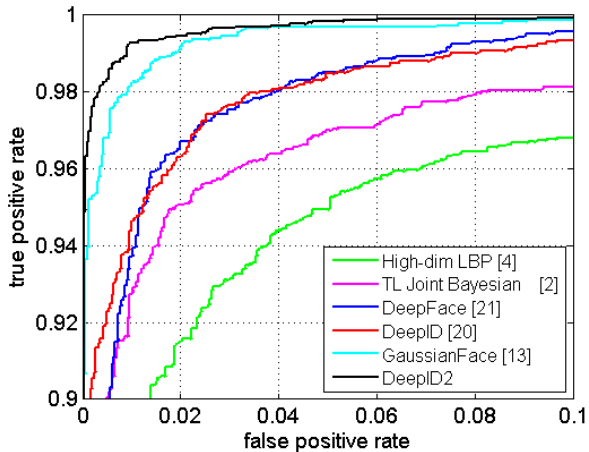

Figure 7: ROC comparison with the previous best results on LFW. Best viewed in color.

To further exploit the rich pool of DeepID2 features extracted from the large number of patches, we repeat the feature selection algorithm for another six times, each time choosing DeepID2 features from the patches that have not been selected by previous feature selection steps. Then we learn the Joint Bayesian model on each of the seven groups of selected features, respectively. We fuse the seven Joint Bayesian scores on each pair of compared faces by further learning an SVM. In this way, we achieve an even higher **99.15**% face verification accuracy. The accuracy and ROC comparison with previous state-of-the-art methods on LFW are shown in Tab. 4 and Fig. 7, respectively. We achieve the best results and improve previous results with a large margin.

## 5  Conclusion

This paper have shown that the effect of the face identification and verification supervisory signals on deep feature representation coincide with the two aspects of constructing ideal features for face recognition, i.e., increasing inter-personal variations and reducing intra-personal variations, and the combination of the two supervisory signals lead to significantly better features than either one of them. When embedding the learned features to the traditional face verification pipeline, we achieved an extremely effective system with **99.15**% face verification accuracy on LFW. The arXiv report of this paper was published in June 2014 [19].

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
