[Reviews · NeurIPS 2014]

Submitted by Assigned_Reviewer_3

This paper's core finding is that combining an identity classification task as well as metric-learning-style verification task helps to learn better features for face classification/verification. The "verification task" here tries to decrease feature-space distance between instances of the same identity, and increase distance between those of different identities. This improvement is embedded in a state-of-the-art system for face verification, which uses convnets trained on many (400) different views to generate features, distilled into a small set of 25 using feature selection. Very good results are obtained and experiments performed using LFW as a test set.

Overall, these are very good results obtained using a somewhat complex pipeline, and a good investigation into the contribution of each "task" in the loss for feature learning. Combining the tasks has been hinted at before (e.g. siamese network after classification pre-training for DeepFace), but this work brings this out specifically. Some of the writing could be a bit less dry (and edited for English), but this does not interfere with the understanding of the paper.

Further comments:

* I don't understand how "m" in eqn 1 was selected. If it is learned as a parameter, it seems it would collapse to m=0, or at most the minimum ||fi-fj|| (even with the procedure indicated at l.152), since this would always give zero Verif error for negative pairs.

* Is there a more concrete reason why including the identification task improves verification performance and/or learning ("richer information" is a bit vague)? E.g. the gradient from the verification task pushes apart just two points, while the softmax grad in the identification task pushes the correct identity away from the classification-layer templates for *all* other identities at the same time.

* It would be nice to see individual performance of the best few selected network regions/views, or an evaluation using 1, 2, 4, ..., 25 views. The final combination achieves good performance, but it would be good to also see the individual contributions.

* l.38 "eternal topic": this is an odd choice of word (it implies the topic will be debated forever, when in fact it seems to be making good progress); a "central topic" may be better here.
Summary: This is a well-described state-of-the-art system, and explicitly explores the effect of using each of two tasks during feature learning. It seems a bit incremental in its contributions, putting together several existing ideas, but I don't see this as a major drawback.

Submitted by Assigned_Reviewer_7

This paper studies the face recognition problem by using a deep learning framework. The key idea is to learn features by using both face identification and verification signals as supervision. This will simultaneously enlarge inter-personal differences and reduce intra-personal variations. Such a motivation is simple and it has been widely studied before, e.g. LDA. Bringing such a technique into the deep learning framework is novel to me, and the obtained 99.15% verification accuracy on LFW is the best result compared with state-of-the-art.

This paper is also very well written. They also provides several experiments to verifiy their idea from different perspectives.
Summary: A novel deep learning frmework by using both face identification and verification signals as supervision is proposed. The best 99.15% verification accuracy on LFW is achived. The idea is simple and the performance is very well.

Submitted by Assigned_Reviewer_21

Summary
The paper proposes a face verification method using deep convolutional neural networks based on identification (e.g., softmax classifier) and verification objectives with landmark-based multi-scale approach. The approach in this paper is very closely related to the work of [18] except 1) the training objective is augmented with verification objectives and 2) the selection of patches (location and scale) proposed by facial landmarks. The model is trained on CelebFaces+ database and evaluated on LFW, achieving state-of-the-art performance on face verification.

Quality
The paper introduces few neat ideas to extend the previous state-of-the-art system on face verification and the experiments are well executed.

Clarity
The paper is clearly written. Here are some comments:
1) I think more head-to-head comparison to the work of [18] is necessary as both follow the almost identical verification pipeline except for details.
2) Details about training m (margin, Equation (1)) is required (e.g., training/updating m doesn’t appear in Table 1.
3) The best verification accuracies in figures and tables (e.g., figure 3, 4, table 2, 3, and 4) do not match. Having more complete captions for tables and figures with details about experimental setting would be helpful.

Originality
Although the techniques are all existing, it doesn’t seem to hurt the paper as it combined the ingredients well and executed to show state-of-the-art performance.

Significance
Face verification is an important problem and deep learning has shown recent success on this problem. This paper further pushes the bar and makes the paper significant. It could be more significant if authors can include the evaluation result on YouTube Face database, which is bit more challenging dataset.
Summary: The paper extends the previous work by augmenting the CNN objective function with verification objectives and adopting better region selection algorithm. The paper demonstrates the state-of-the-art face verification performance on LFW database.
Author Feedback
Author rebuttal: We thank the reviewers for their careful evaluation of our paper.

To R21

Q1. I think more head-to-head comparison to the work of [18] is necessary.

A1. The key difference with [18] is that the objective function of training the network is augmented with verification objectives while [18] only used the identification objective. It brings significant improvement as shown in Fig. 3 and Tab. 2. Another difference is patch selection. Its effect is shown in Tab. 3. More head-to-head comparisons will be added in the final version.

Q2. Details about training m (margin, Eq. (1)) is required (e.g., training/updating m doesn’t appear in Tab. 1).

A2. Learning m is described in lines 151-153. m is not updated by every single training pair according to Eq. (1). Instead, it is updated every N (e.g., N = 10,000) training pairs as a threshold of the feature distances ||fi – fj|| of these training pairs which minimizes their verification errors. More details will be added and Tab. 1 will be updated.

Q3. The best verification accuracies in figures and tables (e.g., Fig. 3, 4, Tab. 2, 3, and 4) do not match.

A3. The best accuracies in Fig. 3, 4, and Tab. 2 are consistent, since they are from experiments using the same face region. The automatically selected face regions in Tab. 3 are different from the region (which is a full face region chosen by us) used in the previous experiments. Therefore accuracies in Tab.3 are different from those in Fig. 3, 4, and Tab. 2. Tab. 4 shows our final result, which is slightly higher than the best accuracy in Tab. 3, by combining features from more face regions.

To R3

Q4. I don't understand how "m" in Eq. (1) was selected. If it is learned as a parameter, it seems it would collapse to m=0, or at most the minimum ||fi-fj|| (even with the procedure indicated at l.152), since this would always give zero Verif error for negative pairs.

A4. Eq. (1) is minimized given a fixed m. m is NOT optimized by minimizing Eq. (1). Instead, it is selected as a threshold of the feature distances to minimize the verification error of the previous N training pairs. Since positive and negative sample pairs are randomly generated with an equal probability, m must be a positive value. Setting m to zero can only correctly classify all the negative pairs and get approximately 50% verification accuracy. In our experiments, m will converge to a positive value.

Q5. Is there a more concrete reason why including the identification task improves verification performance and/or learning ("richer information" is a bit vague)?

A5. The identification signal pulls apart features of different identities by mapping them to different points in high dimension. Each input image is required to be distinguished among approximately 10,000 different identity classes at the same time to guarantee that features of each identity are different from those of all the other identities. So the reviewer’s comment is right, i.e. the correct identity is pushed away from ALL the other identities. Therefore features learned in this way must have diverse identity-related (inter-personal) variations. On the other hand, the verification objective considers sample pairs. When it pulls features of one identity away from features of another identity at each time, it may result in features being closer to those of other identities. Therefore verification is weaker than identification to learn diverse inter-personal variations. Experimental results in Fig. 3 show that the performance drops a lot if only the verification objective is used.

Q6. It would be nice to see individual performance of the best few selected network regions/views, or an evaluation using 1, 2, 4, ..., 25 views. The final combination achieves good performance, but it would be good to also see the individual contributions.

A6. We will include the individual performance. In general, the eye regions are easier to recognize and lead to better performance than those that exclude the eye regions. An evaluation using 1, 2, 4, …, 25 views is shown in Tab. 3.